# Direct evidence for microbial-derived soil organic matter formation and its ecophysiological controls

Cynthia M. Kallenbach[1,2], Serita D. Frey[1] & A. Stuart Grandy[1]

Soil organic matter (SOM) and the carbon and nutrients therein drive fundamental submicron- to global-scale biogeochemical processes and influence carbon-climate feedbacks. Consensus is emerging that microbial materials are an important constituent of stable SOM, and new conceptual and quantitative SOM models are rapidly incorporating this view. However, direct evidence demonstrating that microbial residues account for the chemistry, stability and abundance of SOM is still lacking. Further, emerging models emphasize the stabilization of microbial-derived SOM by abiotic mechanisms, while the effects of microbial physiology on microbial residue production remain unclear. Here we provide the first direct evidence that soil microbes produce chemically diverse, stable SOM. We show that SOM accumulation is driven by distinct microbial communities more so than clay mineralogy, where microbial-derived SOM accumulation is greatest in soils with higher fungal abundances and more efficient microbial biomass production.

[1] Department of Natural Resources and the Environment, University of New Hampshire, Durham, New Hampshire 03824, USA. [2] Soil and Crop Sciences Department, Colorado State University, Fort Collins, Colorado 80523, USA. Correspondence and requests for materials should be addressed to C.M.K. (email: kallenbachcm@gmail.com).

For nearly a century, soil organic matter (SOM) formation in conceptual and quantitative models has been depicted primarily as a function of plant inputs and their chemistry[1–3]. As such, chemically diverse and stable SOM originates from the preservation of biochemically recalcitrant complex plant polymers, such as lignin derivatives and long-chain lipids[2,3]. However, soil microbial communities are adept at decomposing a wide range of plant compounds and using the carbon (C) to synthesize their own biomass. The importance of soil microbes in processing plant inputs and synthesizing SOM is not conceptually new[4–6], though until recently it has largely been overlooked as a primary pathway of SOM formation[7]. In a significant departure from the dominant plant-based models of the past, microbial contributions to SOM formation have gained widespread acceptance, in part due to advances in molecular analytical techniques[1,7,8]. Mounting evidence demonstrates that some decomposition-resistant SOM bears little chemical resemblance to plant material, but is instead characteristic of microbial cells, excretions and cytoplasmic materials[9–11] stabilized via organo-mineral and organo-metal oxide interactions[12,13]. While the evidence to support this alternative pathway of SOM formation is compelling[8–11,14,15], analytical challenges associated with separating direct microbial and plant inputs to SOM create fundamental uncertainties about the degree to which microbial residues contribute to the formation of stable SOM and its characteristic chemical diversity[11,16,17].

Whether plant inputs are first converted to microbial residues before stabilization influences how SOM responds to land use and climate change[1,18,19] as well as how it should be modelled[20] and managed to promote climate change mitigation[21]. Plant residues that accumulate in soil through physical protection (e.g., inside aggregates) or in zones with low biological activity are susceptible to destabilization following disturbances such as cultivation or in response to environmental change (e.g., temperature increases)[19,22]. If, however, plant materials are synthesized into microbial proteins, lipids or polysaccharides, the resulting organo-mineral associations may include ligand bonds or other strong interactions[22] that have lower temperature sensitivity[13,23] and may better withstand perturbations. Despite the potential importance of microbially derived SOM, experimental evidence for microbial contributions to SOM formation is constrained by methods that select for a limited group of microbial biomarkers, such as amino sugars or select lipids[9,21], is typically correlative and inferential[9,21,24,25], or relies on visualization techniques that cannot easily be scaled up to a whole-soil basis[10]. These constraints have hindered our ability to quantitatively determine the importance of microbial-derived compounds as key proximate inputs to SOM, and thus limit our understanding, management and predictions of SOM dynamics.

Further, many newly developed conceptual and quantitative microbial models put considerable emphasis on the abiotic stabilization[20,26,27] of microbial-derived SOM, yet the ecological controls that regulate the transfer of microbial residues to mineral-associated SOM have yet to be resolved[20]. While clay mineralogy is known to regulate microbial-SOM accumulation[22,28], microbial community structure and physiology likely also determine stable SOM accumulation rates due to differences in microbial residue production[15,21,29]. For example, microbial carbon use efficiency (CUE), the amount of C used for microbial growth relative to total C uptake may have a direct impact on microbial residue production[15,21] but can differ across resource gradients[30] and microbial communities[31]. As CUE increases, relatively more substrate-C goes towards biomass synthesis, potentially increasing the amount of residues available for stabilization[21,29]. Substrate chemistry can alter CUE due to its direct effect on cellular metabolism, but may also drive CUE indirectly, by selecting for distinct microbial communities with different prevailing life histories[31]. For instance, highly reduced substrate-C (high free energy) typically promotes higher CUE within a community but may also select for a copiotrophic-dominated community with an inherently lower CUE[31,32]. Thus, microbial community composition, available substrates and CUE are intimately connected, but the influence of their interactions on microbial-SOM formation has not been well characterized[32].

We use model soils to quantitatively assess whether microbial processing of simple C (i.e., low-molecular-weight) substrates alone, in the absence of complex plant compounds, can build significant amounts of chemically diverse, stable SOM. Since our model soils are initially C- and microbe-free, we eliminate the difficulties of isolating microbial residues that occur when using natural soils. We use a gradient of substrate-C inputs to represent different microbial-available energy in order to facilitate the development of diverging microbial communities and physiologies that are hypothesized to influence SOM chemistry and accumulation rates. The substrate gradient includes monomeric and dimeric sugars since they are abundant energy sources for microbial metabolism in natural soils[33], but also because their rapid microbial uptake and intercellular breakdown should leave little to no unaltered substrate in the soil that would interfere with the detection of novel SOM molecules formed during the experiment[33,34]. Further, we include a more recalcitrant lignin monomeric substrate, as well as plant-derived dissolved organic C (DOC), a natural analog to test whether substrate chemical diversity is requisite to generating SOM chemical diversity. We also compare two clay types (kaolinite or montmorillonite) to investigate the effects of clay mineralogy on SOM accumulation relative to microbial communities and substrate chemistry. We characterize the composition of SOM after 18 months of incubation using high-resolution molecular fingerprinting by pyrolysis-gas chromatography/mass spectrometry (py-GC/MS) to establish the chemical fingerprint of newly formed microbial residues (including cell wall and cytoplasmic materials, metabolites and extracellular excretions). Our model soils accrue C concentrations between 1 and 1.4% C, with a chemical diversity and stability characteristic of natural soils. Substrate type has a stronger influence on SOM development than clay mineralogy. However, this effect appears to be an indirect consequence of diverging microbial communities, where different substrates select for distinct microbial communities, with microbial-SOM accumulation being greatest in soils where fungal abundances are highest and microbial biomass production is most efficient.

## Results

**Model soil systems.** We created initially C- and microbe-free model soils to study proximal drivers of SOM development. Model soils were incubated with a natural soil microbial community inoculum and received weekly C additions of glucose, cellobiose, syringol (a lignin monomer) or plant-derived DOC, combined with a nutrient solution for 15 months. Substrate additions were terminated after 15 months, but the incubation continued for an additional 3 months (18 months total) without further C inputs in order to maximize endogenous C recycling.

Following inoculation of model soils, microbial activity was immediately detectable by a measurable $CO_2$ efflux, and by 6 months post-inoculation microbial extracellular enzymatic activities, $CO_2$ respiration rates and living microbial biomass all indicated that an active microbial community had been established in all treatments (Supplementary Fig. 1). Visually, model soils progressively resembled natural soil, with significant colour and structural development (Fig. 1a). The exception to this

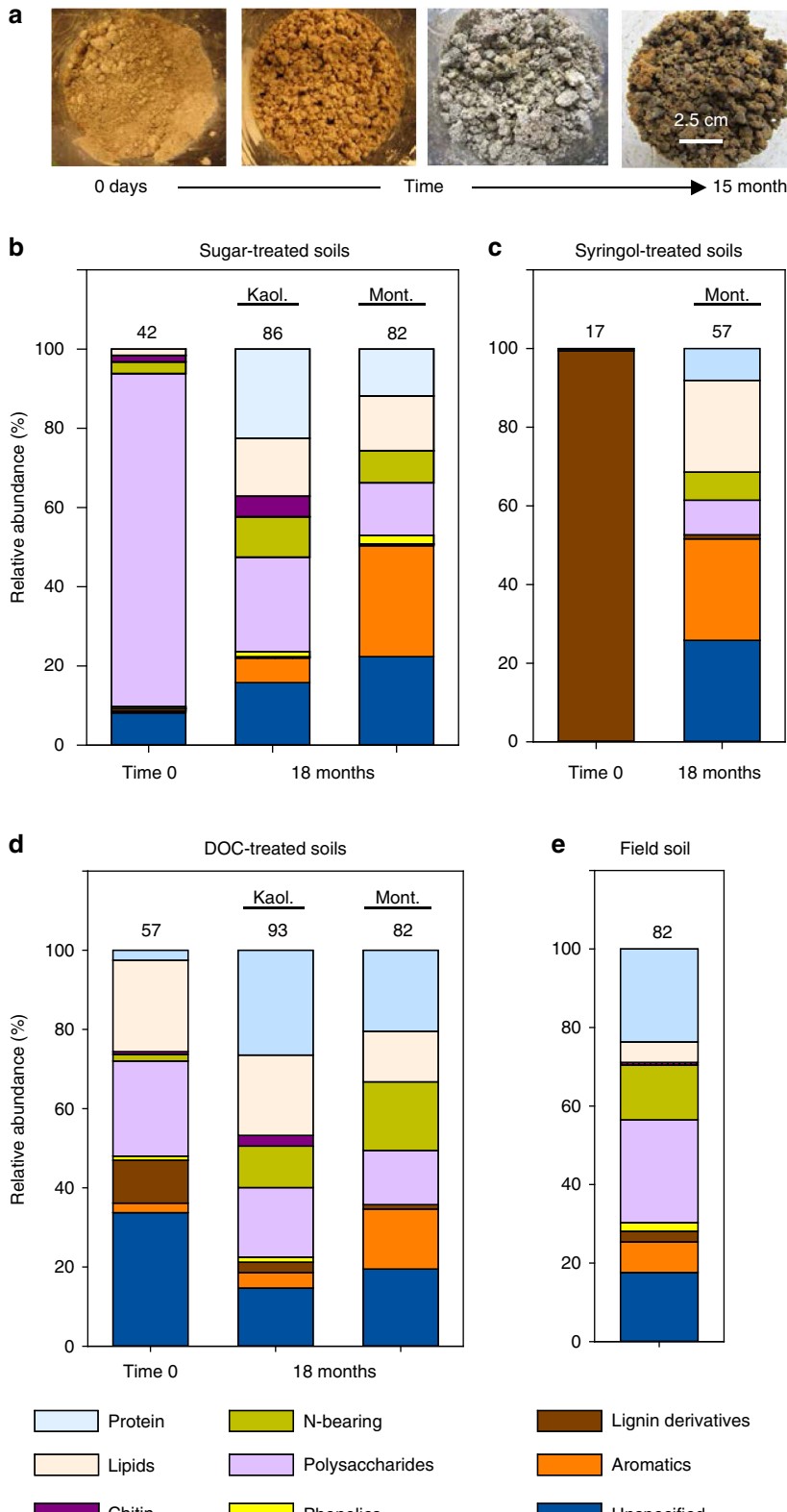

**Figure 1 | Soil development and organic matter chemistry.** Images of sugar-treated model soils over time (**a**); the far left panel is an uninoculated sterile kaolinite and sand mixture, and the far right panel is the same mixture, inoculated and treated with weekly glucose additions for 15 months. Relative abundance of chemical compound groups in substrate (Time 0) and model soils amended with (**b**) sugar, (**c**) syringol and (**d**) plant dissolved organic carbon (DOC). These are compared to soil collected from an agricultural field (**e**). Glucose and cellobiose treatments were averaged since there were no significant differences in their chemistry (ANOVA: $P > 0.05$). Numbers above bars are the total number of identified compounds.

was the kaolinite–syringol treatment, which after the initial 3 months did not exhibit an active microbial community and was thus removed from the study.

*SOM chemistry.* We used py-GC/MS to examine the chemistry of added substrates, along with SOM chemistry at the termination of the study (18 months). Substrate chemistry at Time 0 provides a control to compare changes in SOM chemistry following microbial processing. The visual changes we observed in soil development (Fig. 1a) coincided with the creation of microbial residues, specifically microbial-derived proteins and lipids (Fig. 1b–d). After 18 months, regardless of initial substrate chemistry or clay type, accumulated SOM largely consisted of microbial products (Fig. 1), and the SOM molecular diversity (number of compounds) increased by 54–83% across our model soil systems, with the number of novel compounds varying from 73 to 100% of the total (Supplementary Fig. 2).

Initially, the unprocessed glucose and cellobiose substrates (Time 0) contained primarily polysaccharides (80% relative abundance) while the syringol substrate was largely comprised of lignin derivatives (80% relative abundance) (Fig. 1; Supplementary Note 1). After 18 months, polysaccharide relative abundance in sugar-treated soils declined to 18%. Similarly, in syringol-treated soils, lignin derivatives declined to 1% over that time period. In both the sugar- and syringol-treated soils, the dominant chemical signature of the substrate was replaced by proteins, non-proteinaceous nitrogen compounds and lipids. Lipids and proteins, not initially present in unprocessed substrates, collectively made up 37% relative abundance of all compounds in newly formed SOM (Fig. 1 and Supplementary Table 1). Chitin, originating primarily from fungal cell walls[6], also increased by 5% after 18 months. We also observed increases in both the aromatic and unspecified compound classes after 18 months across all soils. The most abundant compounds within these classes are consistent with abundant compounds observed in fungal biomass derived from natural soil (Supplementary Table 2 and Supplementary Note 1).

Microbial processing of sugars produced SOM similar in chemistry and diversity to soils amended with more recalcitrant syringol, and began to resemble soils treated with more heterogeneous plant-derived DOC, as well as a natural soil (Figs 1 and 2). Further, microbial processing of DOC also increased chemical diversity by 54% (Fig. 1). In contrast to traditional expectations that slowly decomposing plant compounds accumulate in SOM[2,3], lignin derivatives, specific to plant-derived lignin macromolecules, declined from 10 to 2.5% relative abundance in DOC-treated soils, and proteins and lipids increased five-fold ($P < 0.05$).

Significant differences in SOM chemistry due to mineralogy emerged after 18 months (Monte Carlo $P < 0.05$; multi-response permutation procedure $P < 0.05$) (Fig. 2 and Supplementary Tables 1 and 3). For example, kaolinite sugar-treated soils had higher relative abundances of polysaccharides and chitin derivatives and a lower abundance of aromatic compounds compared with montmorillonite ($P < 0.05$). However, there was little influence of the substrate treatment on final SOM chemistry, with, for instance, montmorillonite, sugar- and syringol-treated soils exhibiting similar SOM chemistries at 18 months (Fig. 2).

**SOM accumulation and stability.** We determined soil organic carbon (SOC) accumulation rates and stability in the initially C-free model soils. Across all model soils, SOC increased consistently over time ($P < 0.05$; Table 1). The efficiency with which substrate-C was converted to total SOC—i.e., the amount of SOC remaining relative to the total amount of substrate added—

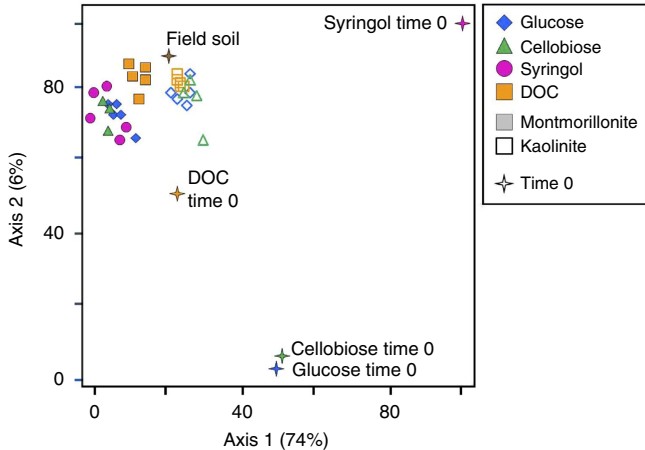

**Figure 2 | Differences in soil organic matter chemistry between substrate and clay types.** Non-metric multidimensional scaling (NMDS) ordination of the relative abundance of chemistry compounds at 18 months for substrate and clay treatments. Open symbols are kaolinite and closed symbols are montmorillonite. For comparison, unprocessed substrates (Time 0) and an agricultural field soil are indicated by a star symbol (Stress = 8.1, Monte Carlo: $P < 0.05$).

declined from an average across all treatments of 32% ± 1.4 at 6 months to 21% ± 0.75 at 18 months Using SOC stocks as an integrator of mass C balance, the majority (>75%) of total substrate-C added was lost via respiration across all treatments by 18 months (Table 1). As a comparison, estimated cumulative respirations represented 50–75% of added C with the exception of DOC-treated soils (Supplementary Table 4). These estimates are based on flux measurements, which exhibited rapid and ephemeral responses to each substrate addition, especially for the monomeric inputs. In DOC-treated soils, cumulative respiration underestimated C loss, compared with calculations using SOC stocks. We attribute this to the more chemically complex DOC treatment having a longer respiration response time that was not completely captured from fluxes collected immediately following substrate additions.

The final total soil C concentrations at the end of the C addition period (1–1.4%) are well within the range of many natural soils[21] (Table 1). The syringol-treated montmorillonite soils accumulated the most C ($P < 0.001$); however, there were no differences in SOC among the glucose- or cellobiose-treated soils in either clay type. DOC-treated soils had higher SOC compared with sugar-treated soils (12.8 mg C per g soil, $P < 0.001$), and also exhibited differences among clay types ($P < 0.05$; Supplementary Table 3).

We determined SOC biological and chemical stability to evaluate its potential long-term persistence. We assessed biological stability by adding a [13]C-labelled substrate mixture (1:1 glutamic acid:glucose at 25 atom % and 50 µg C per g soil) to a subsample of soil from the main experiment and incubating for 3 months (Table 2). The labelled substrate enabled us to use a standard isotope mixing model[35] to determine the amount of previously formed C vulnerable to decomposition by an active microbial community. Chemical stability of accumulated SOC was assessed with an acid hydrolysis fractionation, where the unhydrolysable fraction is considered chemically stable[36]. The majority of SOC was biologically stable (71% ± 2.5; Table 2) and 40% ± 1 was chemically stable to oxidation, which is within the upper range observed for natural soils[36,37]. We observed greater biological stability in montmorillonite than kaolinite model soils only within the cellobiose and DOC treatments ($P < 0.05$; Supplementary Table 3).

**Table 1 | Soil carbon accumulation.**

| Months | Substrate added | SOC (mg C per g soil) | | | | | | | Field soil |
| --- | --- | --- | --- | --- | --- | --- | --- | --- | --- |
| | | Kaolinite | | | Montmorillonite | | | | |
| | | Glucose | Cellobiose | DOC | Glucose | Cellobiose | Syringol | DOC | |
| 6 | 16.80 | 5.10$^{Ab}$ | 5.33$^{Ab}$ | 6.21$^{Bb}$ | 4.01$^{Aa}$ | 3.91$^{Aa}$ | 7.96$^{C}$ | 5.55$^{Ba}$ | 9.18 ± 0.04* |
| 9 | 27.30 | 7.21$^{Ab}$ | 6.55$^{Aa}$ | 8.54$^{Ba}$ | 6.2$^{Aa}$ | 6.72$^{Aa}$ | 14.36$^{C}$ | 8.83$^{Ba}$ | |
| 12 | 37.10 | 8.05$^{Aa}$ | 9.21$^{Ba}$ | 11.10$^{Ca}$ | 7.76$^{Aa}$ | 8.86$^{Aa}$ | 12.98$^{B}$ | 10.73$^{Ba}$ | |
| 15 | 46.90 | 11.08$^{Aa}$ | 11.33$^{Aa}$ | 12.88$^{Bb}$ | 10.43$^{Aa}$ | 10.17$^{Aa}$ | 14.47$^{C}$ | 11.98$^{Ba}$ | |
| 18 | 46.90 | 8.61$^{Aa}$ | 8.88$^{Aa}$ | 10.45$^{Ba}$ | 8.36$^{Aa}$ | 7.98$^{Aa}$ | 13.11$^{B}$ | 11.75$^{Bb}$ | |
| Amount lost | | 38.29 | 38.02 | 36.45 | 38.54 | 38.92 | 33.79 | 35.20 | |
| | | SOC conversion efficiency (mg SOC per g total substrate-C added) | | | | | | | |
| 6 | 16.80 | 0.30$^{Ab}$ | 0.32$^{Ab}$ | 0.37$^{Bb}$ | 0.24$^{Aa}$ | 0.23$^{Aa}$ | 0.47$^{C}$ | 0.33$^{Ba}$ | 0.03–0.33$^{†}$ |
| 9 | 27.30 | 0.26$^{Ab}$ | 0.24$^{Ba}$ | 0.31$^{Ca}$ | 0.23$^{Aa}$ | 0.29$^{Aa}$ | 0.49$^{C}$ | 0.32$^{Ba}$ | |
| 12 | 37.10 | 0.22$^{Aa}$ | 0.25$^{Ba}$ | 0.30$^{Ca}$ | 0.21$^{Aa}$ | 0.24$^{ABa}$ | 0.35$^{B}$ | 0.29$^{Ba}$ | |
| 15 | 46.90 | 0.24$^{Aa}$ | 0.24$^{Aa}$ | 0.27$^{Bb}$ | 0.24$^{Aa}$ | 0.23$^{Aa}$ | 0.47$^{C}$ | 0.26$^{Ba}$ | |
| 18 | 46.90 | 0.18$^{Aa}$ | 0.19$^{Aa}$ | 0.22$^{Ba}$ | 0.18$^{Aa}$ | 0.10$^{Aa}$ | 0.28$^{B}$ | 0.25$^{Bb}$ | |
| Fraction lost | | 0.82 | 0.81 | 0.78 | 0.82 | 0.90 | 0.72 | 0.75 | |

The soil organic carbon (SOC) concentration and conversion efficiency (the proportion of added substrate-C converted to SOC) at 6, 9, 12, 15 and 18 months in soils treated with glucose, cellobiose, syringol or plant leachate DOC. For comparison, field soil results are also presented. Significance among substrates within clay type is indicated by capital letters. Significant pair-wise comparisons among clay types within a substrate group is indicated by lowercase letters (ANOVA: $P < 0.05$) (experimental replication $n = 5$). DOC, dissolved organic C.
*Soybean bulk soils 0–7 cm depth, collected from the W.K. Kellogg Biological Station, Michigan[16].
†Data synthesized from 15 field and laboratory incubation experiments[7].

**Table 2 | Percentage of stable soil carbon.**

| | % Stable C | |
| --- | --- | --- |
| | Chemically stable C | Biologically stable C |
| *Kaolinite* | | |
| Glucose | 37.39$^{Aa}$ | 76.61$^{Aa}$ |
| Cellobiose | 37.37$^{Aa}$ | 63.34$^{Aa}$ |
| DOC | 37.14$^{Aa}$ | 74.13$^{Aa}$ |
| | | |
| *Montmorillonite* | | |
| Glucose | 44.12$^{ABa}$ | 82.76$^{Aa}$ |
| Cellobiose | 36.33$^{Aa}$ | 87.65$^{Bb}$ |
| Syringol | 48.03$^{Ba}$ | 93.42$^{C}$ |
| DOC | 38.65$^{Aa}$ | 93.00$^{Cb}$ |
| *Field soil* | 32–66* | 77.60 ± 3$^{†}$ |

Chemically stable SOC is the per cent of non-hydrolysable C, and biologically stable SOC is the per cent of previously accumulated SOC not mineralized during a 3-month incubation at 6 months. For comparison, natural field soil results are also presented. Significance among substrates within clay type is indicated by capital letters. Significant pair-wise comparisons among clay types within a substrate group are indicated by lowercase letters (ANOVA: $P < 0.05$) (experimental replication $n = 5$). DOC, dissolved organic C.
*Range is the acid unhydrolysable fraction using 6 M HCl from 22 soils from cultivated and grassland soils at depths from 0 to 20, 25 to 50 and 50 to 100 cm (ref. 25).
†Per cent of non-mineralizeable SOC from a 588 day laboratory incubation on eight cultivated and native grassland soils[24].

**Microbial community composition and physiology.** Given its potential influence on SOM accumulation rates and chemistry, we examined microbial community composition using phospholipid fatty acid (PLFA) biomarkers. We observed significant differences in broad microbial groups due to substrate treatment ($P < 0.05$; Supplementary Table 3 and Supplementary Fig. 3). Montmorillonite, syringol-treated soils had lower Gram-negative and Gram-positive bacteria and higher fungal relative abundance compared with sugar-treated soils ($P < 0.001$; Fig. 3). We also observed an effect of mineralogy on the microbial community; fungal abundances were generally higher in kaolinite than in montmorillonite soils, while Gram-negative bacteria were more abundant in montmorillonite soils ($P < 0.001$).

We evaluated community-level CUE to determine whether the distinct community composition we observed between substrates was related to differences in microbial physiology and subsequently SOC accumulation. We estimated the CUE using [13]C-labelled glutamic acid uptake into microbial biomass and respired [13]C-$CO_2$ as a proxy for determining the proportion of new C inputs used for microbial biomass synthesis[30,32]. There was a strong effect of substrate type, where syringol-treated soils exhibited the highest CUE ($P < 0.001$; Fig. 3). Microbial community composition, CUE and SOC accumulation were highly correlated across substrate treatments (Fig. 4). Soils with the highest SOC accumulation exhibited higher fungal relative abundances and CUE (Fig. 4a,c,d). Accordingly, higher fungal abundance was also positively correlated with CUE (Fig. 4b). Though syringol-treated soils exhibited the highest CUE, fungal relative abundances and SOC, the relationships among these variables remain when we analysed only sugar-treated soils (Supplementary Fig. 4). We also observed that soils with higher SOC concentrations had greater lipid relative abundances and lower protein abundances (Fig. 4a).

**Discussion**

Model soil systems provide a platform for directly manipulating specific abiotic and biotic controls on soil processes. They allow for insight into fundamental questions about soil aggregate development[38], organic matter turnover[39] and mineralogical influences on microbial communities and decomposition[40–42]. Here we use model soils for real-time monitoring of microbial-SOM formation and demonstrate that microbial processing of simple C substrates produced an abundance of stable, chemically diverse SOM dominated by microbial proteins and lipids, comparable to natural soils. However, clay mineralogy, a well-known control over SOM dynamics[13,28], had little effect on SOM abundance and stability. Instead the microbial community and its associated physiology was a stronger driver of SOM development. This suggests that the community's physiology and their residue inputs to SOM may be as or more important than the more widely recognized effects of soil mineral structure on soil C stabilization.

Considerable SOM accumulated (8–13 mg C per g soil) within 18 months in our model soils (Table 1). The high microbial activity (Supplementary Fig. 1), cumulative respiration (Supplementary Table 4) and especially SOM chemistry (Fig. 1) in our model soils indicate that substrates were first processed by microbes before becoming SOM. Further, consistent with studies

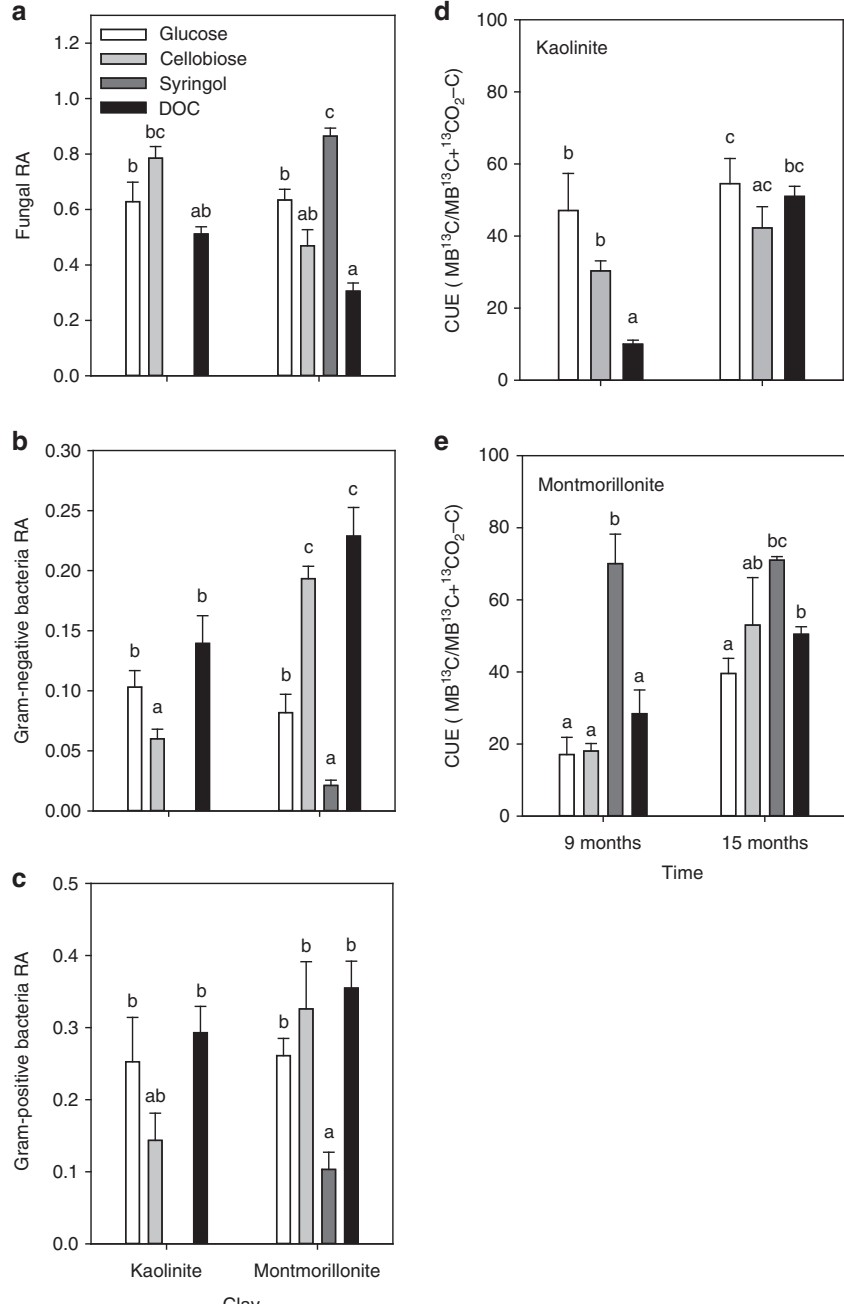

**Figure 3 | Influence of substrate and clay type on soil microbial communities and microbial growth efficiency.** Substrate treatment differences in the relative abundance (RA) of fungi (**a**), Gram-negative bacteria (**b**), Gram-positive bacteria (**c**) at 15 months, and microbial carbon use efficiency (CUE) for kaolinite (**d**) and montmorillonite (**e**) model soils. The CUE values (**d,e**) are for 9 and 15 months of incubation. Significant substrate treatment effects are within clay type or within time and are indicated by different letters (ANOVA: $P < 0.05$). Error bars represent one s.e. (experimental replication $n = 5$).

showing rapid microbial assimilation of low molecular weight compounds[33], including glucose[34,43,44], we found that the three most abundant compounds in unprocessed sugar and syringol declined by 87 and 95%, respectively, following incubation. Moreover, the protein, lipid and chitin classes derived exclusively from microbes constituted 34% of the SOM compound relative abundance across all samples. Many of the compounds not derived from lipids or proteins may also be derived from microbial residues. For example, we know that microbial residues generate a variety of extracellular polysaccharides, and peptidoglycan is a primary cell wall building block[6,43]. We also found that the unspecified compounds toluene, cyclooctatetraene

and the aromatic *m*-xylene are the most abundant compounds within their class in the sugar- and syringol-treated soils, as well as in a separate analysis of natural soil fungal biomass (Supplementary Table 2). It is possible that some SOM was derived directly from added substrates, yet mineral sorption of monomeric compounds has largely been attributed to carboxyl groups that are not present in our substrates[45]. Instead, our data point to SOM originating from microbes, *per se*, rather than the substrates they utilized.

The high molecular diversity characteristic of SOM is often attributed to the chemistry of plant-derived compounds and their decomposition byproducts[46–48]. Whether microbial metabolites,

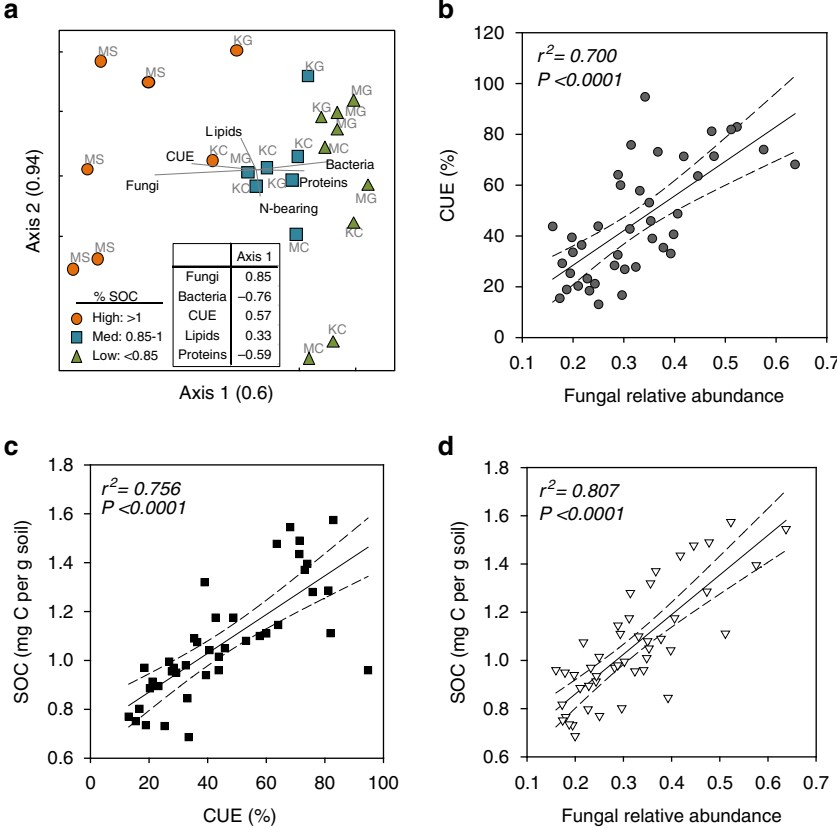

**Figure 4 | Relationships between soil organic matter and microbial variables.** Non-metric multidimensional scaling (NMDS) ordination of accumulated SOC at 12, 15 and 18 months for sugar- and syringol-treated soils (**a**) and Pearson correlations for fungal relative abundances and microbial carbon use efficiency (CUE) (**b**), SOC and CUE (**c**), and fungal relative abundances and SOC (**d**) at 15 months. NMDS points are categorized *post hoc* based on soil C concentrations at 18 months: high SOC (closed circles), medium SOC (closed squares) and low SOC (closed triangles) (Stress = 4.2, Monte Carlo: $P = 0.019$, multi-response permutation procedure for SOC groups: $T = -10.037$, $A = 0,469$, $P < 0.0001$). NMDS table inset is Pearson correlation r values with ordination Axis 1.

secretions and cellular material can also produce chemically heterogeneous SOM remains unclear[48,49]. Our results show that microbial residues can generate SOM chemical diversity. We demonstrate that even in the absence of chemically diverse plant inputs, our model sugar- and syringol-treated soils have a molecular composition and diversity resembling both soils treated with more heterogeneous plant inputs (DOC) and natural field soils (Fig. 1). This highlights the influence of microbial processing on SOM chemistry observed in soils under complex plant communities[49–51], but also that SOM chemical heterogeneity can originate from microbial metabolism of relatively simple C substrates such as root exudates or the soluble components in leaf litter. Certainly, lipids, aromatics and other plant-derived compounds may contribute directly to SOM, but we show that microbial byproducts are also important drivers of SOM chemical heterogeneity.

Changes in SOM chemical composition influence its stabilization potential, interaction with clay minerals and potential responses to anthropogenic disturbances[19,21,23]. Thus, the differences in SOM composition that emerged due to mineralogy could have long-term C-cycling consequences. The underlying cause of these differences is unclear since they could be a direct function of the clay type or due to differences in the chemistry of microbial-derived compounds. Variations in SOM chemistry with mineralogy have previously been attributed to differences in dominant binding mechanisms and selective adsorption among mineral types[52,53]. However, such differences can also result from divergent microbial communities[40,41], and thus the production of

chemically different microbial SOM compounds[43]. Indeed, in kaolinite soil both fungal and lipid abundances were greater relative to montmorillonite soil. Thus, the distinct microbial communities in kaolinite and montmorillonite and the different surface characteristics of these minerals may together explain our observed differences in SOM chemistry. While recent work deemphasizes the importance of plant and microbial chemistry, *per se*, as a direct control over decomposition[54,55], it is an important factor in the formation of organo-mineral interactions[56]. As microbial communities and mineral types change across soils, we would expect that the interaction between microbial input chemistry and mineral surface properties will influence SOM dynamics.

The amount of microbial residues available for mineral stabilization also influences SOM dynamics, and may be regulated by the efficiency of microbial biomass production[14,15,21,29]. Increasingly, CUE is being explicitly represented in models on the premise that a greater proportion of C synthesized into biomass should lead to more microbial residues[20,26,57]. Yet, limited empirical evidence directly linking CUE with microbial residues and SOC concentrations continues to constrain model development[57]. To address this, we examined relationships between SOC concentrations, microbial community structure and variation in microbial CUE.

We observed that soils with the highest SOC concentrations were associated with the highest CUE and fungal abundances (Figs 3 and 4). The influence of substrate on SOM accumulation is not likely a direct function of substrate chemistry, since SOM

chemistry was similar between substrate treatments. Rather, our substrate amendments may be affecting microbial physiologies and consequently microbial residue inputs. The divergent communities between sugar- and syringol-treated soils likely reflect repeated exposure to different substrates selecting for distinct microbial communities with different CUE (Fig. 4a).

Syringol-treated soils consistently exhibited the highest CUE in our experiment (Fig. 3). This contradicts previous arguments that an environment with more labile C is expected to increase CUE and consequently generate more microbial-derived SOM[15,21,30]. However, resource chemistry is not the only factor that influences CUE; distinct microbial communities may have intrinsically different CUEs that correspond with prevailing life-history traits[31]. Based on our PLFA analysis, syringol additions selected for greater fungal abundances, which correlates with higher CUE. Thus, the apparent substrate chemistry may be less important for SOM accumulation than how it influences fungal abundance and microbial CUE long term.

While our approach uses a model system without environmental constraints on microbial biomass production such as predation and nutrient limitations present in natural soil, it provides direct evidence that microbial biomass can produce an abundance of stable, chemically diverse SOM. Moreover, we are able to identify the microbial community characteristics in these systems such as fungal abundances and CUE that influence SOM formation. Our results point to new approaches to rebuild soil C contents that emphasize the influence of substrate–microbe interactions on the synthesis of novel SOM constituents that become mineral-stabilized, and will also inform the necessary validation and parameterization of emerging microbial-explicit C models that stress the importance of microbial residues in SOM dynamics[20,57]. Most of our contemporary predictive models and management strategies for SOC are still based on traditional concepts of plant chemical recalcitrance, yet the processes that govern microbial-SOM accumulation and stabilization likely differ from those of plant-derived SOM[18,19,22]. For example, the effects of elevated temperatures and soil disturbance may stimulate the decomposition of plant-stabilized SOM but the consequences for microbial-stabilized residues are less clear. It is likely that environmental changes such as soil warming influence microbial-SOM, not necessarily by destabilizing it, but via alterations in the capacity of the microbial community for efficient residue production[30]. Moreover, our data provide critical empirical insights to advance new microbial-explicit biogeochemistry models, including how microbe–substrate interactions influence SOM dynamics. Thus, our results provide new perspectives on the origin of stable SOM that will help foster the progress needed to model and manage soils to promote enhanced SOM accumulation.

## Methods

**Experimental conditions and sampling.** Laboratory mesocosms consisted of 100 g dry wt of 33% kaolinite or montmorillonite clay mixed with quartz sand. The cation exchange capacity was 5.2 and 28.6 meg per 100 g soil for the kaolinite and the montmorillonite soil mixtures, respectively. Metal concentrations for the kaolinite mixture were 5.6% Al, 1% K, 0.9% Fe, 0.14% Ca, 0.0.11% Na and 0.06% Mg. Montmorillonite mixture metal concentrations were 2.2% Al, 0.8% K, 2.3% Fe, 0.39% Ca, 0.14% Na and 0.32% Mg. Soil mixtures received weekly additions of glucose, cellobiose, syringol or switchgrass plant DOC (0.7 mg C per g dry soil) and biweekly additions of a multi-nutrient Hoagland's solution (0.023 mg N per g dry soil). The substrate-C (5 ml of 14,000 ppm C) and nutrient additions (0.5 ml of 4,666 ppm N) were syringe-injected throughout the model soils to facilitate uniform substrate application. The rate of weekly C additions reflects the upper range of natural C inputs to soil from daily root exudations. Total C inputs from added substrates over the course of 15 months amounted to 46.9 mg C per g soil and total inorganic N from nutrient solution was 1.56 mg N per g soil. To create DOC, we mixed 150 g switchgrass (*Panicum virgatum*) tissue (leaf, stem and seed) with 2 l H$_2$O and shook for 24 h before filtering. The filtrate was then partially evaporated until the C concentration was 18 mg C per ml (C:N 20). We created the microbial

soil inoculum in a soil slurry (1:100 w/v) using soil collected from a Michigan bromegrass (*Bromus inermis*) field at 0–7 cm depth. We then syringe-injected 100 μl of inoculum solution (∼0.00001 g natural soil per g model soil) into the soil mixture. Each soil mesocosm treatment was replicated five times with five destructive harvests at 6, 9, 12, 15 and 18 months. During incubation, soils were maintained at 25 °C, 45% water holding capacity, and a pH between 4.6 and 5. The pH of DOC-treated soils, however, increased to 7 after a few months of incubation and therefore DOC biological data are minimally included in our analyses. During the 15-month incubation period no soil mixing occurred in order to better simulate an undisturbed soil environment and to minimize aggregate destruction.

After 15 months, substrate and nutrient additions ceased but soils were maintained for an additional 3 months under the previous incubation conditions with periodic hand mixing to facilitate microbial uptake of any remaining substrate or unstabilized microbial biomass. Microbial community activity was monitored weekly by CO$_2$ flux measurements for the first 9 months and then biweekly thereafter, and by microbial hydrolytic and oxidative potential enzyme activities[58] at each mesocosm harvest.

**SOM chemistry and concentrations.** We assessed SOM chemistry using pyrolysis-GC/MS[59]. Compound peaks were analysed and identified with the Automated Mass Spectral Deconvolution and Identification Systems (AMDIS V 2.65) software, the National Institute of Standards and Technology (NIST) compound library and published literature[59]. Organic matter compounds are expressed as the % relative abundance of total sample peak area and classified based on origin (lipids, lignin derivatives, polysaccharides, proteins, non-protein N-bearing and phenolics). Identifiable compounds with potentially multiple origins were classified as unspecified. Aromatic compounds also with an unspecified origin were classified as aromatics. The SOM chemical diversity was determined based on the total number of identifiable compounds that were >1% relative abundance.

Total SOC concentrations were determined on an elemental analyser (Costech ECS 4010). Total microbial biomass carbon (MBC) was determined by the difference in C concentration (using a Shimadzu TOC-L analyser) in chloroform fumigated and unfumigated soil subsamples extracted with 0.5 M K$_2$SO$_4$. We calculated the total SOM conversion efficiency as the amount of total SOC relative to the total substrate-C added for each destructible harvest, where the total substrate added is the sum of the weekly substrate addition within the time period up until harvest.

**SOM stability.** We determined biological stability in a separate 3-month incubation on soil subsamples harvested at 6 months. During this separate incubation experiment, soils received three applications of $^{13}$C-labelled 1:1 glutamic acid/glucose 25 atom % mixture at 50 μg C per g soil. The labelled substrate enabled us to use a standard isotope mixing model[35] to determine the amount of newly formed C vulnerable to decomposition by an active microbial community. During the 3-month incubation, soils were frequently hand-mixed to facilitate SOM mineralization and maintained under the same incubation conditions as the long-term incubation. We also assessed the chemical stability of the accumulated SOM with acid hydrolysis fractionation[37]. After the full 18 months incubation, 1 g soil was treated with 50 ml 6 M HCl and heated at 100 °C for 18 h. After heating, the acid hydrolysable fraction was decanted and the remaining non-acid hydrolysable (i.e., stable) residue was rinsed four times with DI-H$_2$O and isolated by centrifugation[36,37]. The chemically stable C was then determined based on the differences in C remaining in the non-acid hydrolysable residues and the initial C content of the sample.

**Microbial community composition.** We determined differences in microbial community composition from PLFA biomarkers, identified and quantified at 12 and 15 months[60]. We included a phosphate buffer in the single-phase solvent system (chloroform) to extract only viable microorganisms. Lipid extracts were fractionated on silicic acid columns to isolate and collect polar lipids. Polar lipids were methylated to fatty acid methyl esters with 0.2 M methanolic KOH, purified and then analysed on a Varian 3800 GC-FID. Peaks were quantified using internal standards and identification was based on retention time data with known standards. Fungal relative abundance was determined by the sum of polyenoic unsaturated fatty acids (18:2ω6 and 18:1ω9c), Gram-positive relative abundances by the sum of the total branched, saturated fatty acids (i15:0, a15:0, i16:0, i17:0 and a17:0) and Gram-negative relative abundances by the sum of monoenoic and cyclopropane unsaturated fatty acids (16:1ω7c, 16:1ω7t cy17:0, 18:1ω7c and cy19:0). We did not detect any actinobacterial fatty acids.

**Soil microbial CUE and enzyme activities.** We characterized microbial CUE at 9 and 15 months by measuring the amount of a $^{13}$C-labelled glutamic acid incorporated into MBC and respired as $^{13}$CO$_2$–C (refs 21,30). Fresh soil subsamples were amended with 25 atom% labelled $^{13}$C glutamic acid (50 μg C per g dry soil, <1% total soil C) and incubated at 25 °C for 22 h. The 22-h incubation time was based on preliminary respiration curves to determine the period when the majority of glutamic acid is utilized but before substrate recycling begins. Following incubation, soils were extracted for $^{13}$C-glutamic acid incorporation into MBC by extracting and analysing MBC as discussed previously and determining its isotopic

composition on a 1030 TOC Analyzer (OI Analytical, College Station, TX) interfaced with a PDZ Europa 20–20 isotope ratio mass spectrometer. A 12-ml $CO_2$ sample was also collected for determining $^{13}CO_2$–C respiration on a GC-isotope ratio mass spectrometer (ThermoScientific, Bremen, DE).

This approach estimates the amount of C allocated to biomass per unit of C substrate consumed and represents a proxy for microbial CUE. We calculated CUE as $[MB^{13}C/(MB^{13}C + {}^{13}CO_2-C) \times 100]$, where $MB^{13}C$ and $^{13}CO_2$-C were determined using a standard isotope mixing model equation[29] and represent the amount of substrate incorporated into MBC and the substrate-C respired as $CO_2$, respectively. We used glutamic acid to estimate microbial community CUE across all substrate-treated soils since it is taken up directly into microbial cells. This approach allows us to more directly compare CUEs across treatments since substrate-C allocation towards enzyme production is minimized and we can better approximate the efficiency of new biomass growth rather than substrate degradation efficiency[32] (Supplementary Note 2).

The potential activities of microbial extracellular enzymes were measured at 6, 9, 12 and 15 months of incubation. We used a standard fluorometric method to assess hydrolytic enzyme activity by adding 1 g soil dry weight homogenized in a slurry with 50 mM sodium acetate buffer[58]. The buffer solution was adjusted to the average soil pH of 4.9. Soil homogenates were added to black, 96-well microplates with compound-specific fluorescing substrates bound to methylumbelliferone. Oxidative enzyme activity (phenol oxidase and peroxidase), associated with lignin breakdown, was measured spectrophotometrically using clear 96-well microplates.

**Statistical and data analyses.** Data were analysed using a linear mixed-model two-way analysis of variance (ANOVA) to determine differences in SOM chemistry, SOC concentration, SOC conversion efficiency, SOC stability, PLFAs and CUE between clay mineralogy and substrate type. Treatment replicate ($n = 5$) was used as a random effect, and clay and substrate were treated as fixed effects. Since the experimental design was unbalanced (there was no syringol treatment within kaolinite soils), analyses for differences in main effects of substrate and mineralogy and their interactions did not include syringol. A one-way ANOVA was used to examine the effect of substrate within a clay treatment, allowing us to include syringol-treated soils in our comparisons. In many analyses the glucose and cellobiose treatments were not statistically different ($P > 0.05$). In these cases, glucose and cellobiose data were averaged and reported as a 'sugar' treatment. Pair-wise comparisons between treatments were determined by Tukey's HSD ($P < 0.05$). We used Pearson's correlation analysis to examine relationships between SOC concentrations, microbial CUE, microbial community composition and SOM chemistry. All ANOVA and correlation analyses were performed in SAS v.9.3 (SAS Institute, 1999) using PROC MIXED and PROC CORR. When necessary, data were normalized using log transformation before analysis of variance analyses. Significance for all analyses was determined at a probability level of $P < 0.05$ unless otherwise stated. Variation around means are reported or shown as standard error.

We also used non-metric multidimensional scaling (NMDS) (PC-ORD; version 4.14) to explore differences in SOM chemistry and community composition. Pyrolysis-gas chromatography/mass spectrometry compound relative abundances and PLFA biomarkers were relativized to maximum and used separately in NMDS matrices. We also used NMDS to determine relevant parameters influencing SOC concentrations for use in our correlation analyses, where SOC concentrations at 9, 12, 15 and 18 months were grouped a priori into low, medium and high SOC concentrations. The Sorensen (Bray-Curtis) index was used as a distance measure for all ordinations. Final NMDS solutions were considered acceptable if Monte Carlo simulations had stress values < 20 and a solution stability of < 0.005. We determined significant groupings by the Monte Carlo stress reduction ($P < 0.05$) and multi-response permutation procedure ($P < 0.05$).

**Data availability.** The data that support the findings of this study are available on request from the corresponding author (C.M.K.).

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

## Acknowledgements

This work was supported by the National Science Foundation Doctoral Dissertation Improvement Grant to C.M.K. (DEB-1311501). Partial funding for this research was also provided by the Department of Energy's Great Lakes Bioenergy Research Center (DOE Office of Science, DE-FC02-07ER64494 and Office of Energy Efficiency and Renewal, DE-ACO5-76RL01830) and by the New Hampshire Agricultural Experiment Station (scientific contribution number: 2605; USDA-NIFA Hatch project: 230340). We are grateful to Cory Cleveland and Phil Robertson for providing comments on previous manuscript drafts.Disclaimer

Any opinions, findings and conclusions or recommendations expressed in this material are those of the authors and do not necessarily reflect the views of the National Science Foundation or the DOE.

## Author contributions

C.M.K. and A.S.G. conceptualized the study. C.M.K. collected the data and performed all the statistical analyses. All authors contributed to manuscript development and revisions.

## Additional information

**Competing financial interests:** The authors declare no competing financial interests.

