## [Peer Review File · Nature Communications]

Reviewers' comments:

Reviewer #2 (Remarks to the Author):

I found this to be an exceptionally clearly written manuscript that describes a simple, elegant experiment to test the importance of microbial growth (and death) on the generation of soil organic matter. One glaring omission is the comparison of this study to other recent work that examines the same idea using artificial soils (Hemkemeyer et al. (2014) FEMS Microbiol Ecol 90: 770-782; Vogel et al. (2014) Soil Biology & Biochemistry 79: 57-67; and the references cited in these relating to prior artificial soil studies).

I. 56-Although currently in vogue, the funneling of plant input through the microbial biomass is really not "new". Jenkinson (1977) famously stated that the microbial biomass is the needle through which all plant material much pass. Similarly, studies in the 60s and 70s showed that newly synthesized microbial biomass was found in the then-accepted solubility fractions used to describe soil organic matter (J.P. Martin, Roger Knowles, etc.). Like many ideas in science, this one is spiraling around again, albeit with better techniques, more clever experiments, etc. I would agree that it has been popularized and is the current favorite paradigm.

I. 197-My preference would be to not introduce RA as an abbreviation.

I. 146-PLFA do give a broad fingerprint and have fungal and bacterial markers; however, it would have been desirable to have some deeper amplicon sequencing done on these samples to see if key taxa might be identified. (I realize that asking more is not quite "fair", so this should not be considered a major flaw.)

I. 200-If I did my calculation correct, your mesocosms held 100 g "soil". It might be helpful to put this detail somewhere in the methods.

I. 201-Wasn't there also a destructive sampling at 18 mo?

I. 207-So, no mixing in prior 15 mo, correct? Why not?

I. 224-"Shimadzu"

I. 258-How do these values compare to your mass balance approach (Table 1), which I think would be more compelling data?

I. 281-"different" rather than "significant"

I. 339-"regular"

Reviewer #3 (Remarks to the Author):

The manuscript describes a study on soil organic matter formation by microbial communities in an incubation experiment. The manuscript is well written and the research questions are well-formulated and relevant in the field of soil science.

The experimental approach is good and the data presented here together with those included in the supplementary information show a thorough characterization of the relevant microbial and carbon degradation processes. If the concerns listed below are addressed, it would make a valuable contribution to our current knowledge.

Some of the choices in data representation and interpretation are unclear, or could be reasoned better. Furthermore, the discussion of the results lacks context of relevant previous studies that have been done on this topic, or with similar experimental approaches. In particular, similar incubation experiments on the effect of clay mineralogy on microbial community establishment (e.g. Ding et al. 2013, Babin et al. 2013) and OM turnover (e.g. Pronk et al. 2012 and 2013, Saidy et al. 2013, Kaiser et al., 2014) should be addressed.

It is confounding how the authors use the term "soil" in the manuscript. E.g. in line 36 it is mentioned that you incubated "initially C free soil" or in line 85 you mention "soil mecososms" (e.g. line 85). I

propose to make it clear that not a soil was used, but a mixture of clay mineral and quartz sand.

Data and data evaluation: Essential information is missing

1. What is the total amount of substrate-C and N added to the mineral components/mixtures (kaolinite, montmorillonite, sand)?

What are the total amounts of CO₂ respired after 6, 9, 12, 15 and 18 months?

A detailed balance is needed.

Not all "SOM" after 18 months (Fig.2) can be considered as microbial-derived. Some SOM may be the "left over" of your substrate additions e.g. substrates and Hoagland solution and DOC that reacted e.g. with mineral surfaces.

How can the authors be sure that it is all microbial necromass? What about the contribution of microbial extracellular polymers (EPS). This cannot be ruled out with the data provided and might be a relevant proportion of the OM accumulated.

Consequently the values in Table 1 are maximum values for "microbial derived SOM" and for the "SOC-conversion efficiency" under the specific conditions.

I propose to discuss the absolute values for "microbial derived SOM" and for the "SOC-conversion efficiency" more cautiously. To evaluate the SOM conversion efficiency it would be very helpful to present a balance that includes C losses by respiration.

2. CO₂ mineralisation rates presented in Extended Data Figure 1 can also be used to discuss the biological stability of the substrate and SOM in the artificial soils.

3. Why didn't you calculate CUE directly by using microbial biomass C and CO₂ data that are presented in Extended Data Figure 1? Wouldn't this be a more direct proxy for the CUE of the microbial biomass that is actually active in the artificial soil amended with the specific substrates. There may be substrate recycling during the incubation but what does it matter if original or recycled substrate is used by the microorganisms to obtain a proxy for the CUE of the actual microbial biomass?

As stated in lines 75ff, CUE is influenced strongly by the substrate chemistry. For your CUE estimation, you used glutamic acid as substrate with a very narrow C/N ratio that is very different to the C/N ratio of the mixture of Hoagland's solution you used and your substrates or your DOC solution. You should discuss your CUE values against this background. Explain why you have chosen glutamic acid as substrate? What can be concluded e.g. from the result, that a microbial community that is adapted to mineralize syringol can use glutamic acid very efficiently?

Regarding the MBC and CO₂ respiration rates in Extended Data Figure 1, I would expect a lower CUE in the syringol treatment compared to more labile C-amendments. This tendency is contrary to your result (see line 163 ff) but in accordance to results from existing literature.

Line 120-127:

I agree with the authors that microbial processing is highly important for SOM formation. However, I question the reasoning the authors present here that their experiment demonstrates that the overriding effect is microbial processing. The material found after termination of the experiment can only be derived from microbial processing of the simple substrates added.

The incubations carried out here were done using relatively simple OM compounds. Even the DOC used was a switchgrass extract, not taking into account other sources of plant OM. It is not very surprising that the number of compound present increased after microbial processing for 18 months. It would also be interesting if the authors compared the OM compositions of their samples after incubation with that of microbial biomass. This may give more indication on the type of compounds preserved in soil.

Lines 54-58:

How can this be a "new pathway of SOM formation" when the input composition of microbial necromass to SOM formation is described in detail in a review in 2002 (Kogel-Knabner, 2002)?

Line 138-140:

The differences between clay mineral composition are quickly discounted. However, looking at figure 2 there seem to be consistently higher aromatics contents in the montmorillonite treatment. There also seem to be differences in MBC and CO₂ (SI 1) and the NMDS points of the SOM chemistry cluster separately (SI 3). It would be valuable if the authors address these differences in the text.

Line 158-170 and figure 3

This reasoning here is difficult to follow and seems somewhat circular. If fungal relative abundance correlates to SOC and CUE, it is a given that SOC also correlates to CUE (figure 3). Furthermore, the relative abundance of fungi, lipids and CUE are clearly not independent variables in this experiment. Particularly, since the difference seem to be dominated by the syringol treatment. How much of this can be explained by a higher capability of fungi to degrade the aromatic ring structure of syringol versus bacteria?

Line 170: "...characteristics that are highly sensitive to land use and climate change" this statement seems to come out of nowhere. This study seems to be more about the type of OM input than about climate change.

Minor comments

- Figure 1: I am not sure this provides useful information for the reader
- Figure 2: Should it be called lignin-derivatives if not lignin but specific compounds (i.e. syringol) were added?
- Figure 2: Improve contrast. The difference between lignin derivatives and chitin, and phenolics and proteins is difficult to see when printed

Reviewers' comments:

Reviewer #2 (Remarks to the Author):

Review response comments:

The authors have responded thoroughly to the comments of the two reviewers. I am satisfied with their response except in one area: use of cumulative CO₂ data to close the mass balance. Despite the argument provided by the authors, the sampling intensity (about 54 times during about 540 days) is reasonably intense and I think sufficient to at least calculate a ballpark estimate of the cumulative amount of CO₂ produced at the times that soils were destructively sampled. I think this should be done and the data added to Table 1. It is not important that the balance close perfectly, but if it is off by more than 10% or so, and especially if there is a consistent over- or under-estimate, then it may suggest that the data are not as sound as they should be.

Specific points:

l. 118-136—This is a stylistic comment, but the first two paragraphs of the results are really a recapitulation of the methods. But maybe it has to be this way because of the structure of sections in this journal.

l. 162—Is this really “chitin” or would bacterial cell wall materials also be lumped into this category?

l. 352—“months”

l. 432—The statistical analysis description suggests that interactions between substrate and clay types were calculated, but it is unclear if they were ever significant. The presentation of the data in the tables and figures is not clear enough to determine if there were interactions or not, although given the unbalanced design they seem to be presented as if there were interactions.

Table 1—Show all data to hundredths position. Also, the use of significance letters is confusing. Usually the alphabetical order follows either the increase (or decrease) of the values, but this is inconsistent (e.g., 6 mo, substrate comparisons for montmorillonite treatment—why does C represent a number between those associated with A or B?). This point about significance letters is also the case in the other tables and figures.

Reviewer #3 (Remarks to the Author):

The authors have positively responded to the points raised in the review and amended the manuscript. They have now also included further references that put their work now in the correct context with respect to previous attempts to investigate the microbial contribution to soil organic matter formation.

One point still remains: The term “microbial residues” used now, instead of “microbial necromass” is misleading and often the terms are used in the literature as synonym. It is thus necessary to give a clear definition, what the term encompasses. It is also necessary to state that both necromass *sensu strictu* as well as extracellular polymeric substances (often called EPS) cannot be differentiated with the present experiment. EPS are released from living cells and it should be made clear what materials are dealt with in the paper.

REVIEWERS' COMMENTS:

Reviewer #2 (Remarks to the Author):

Thank you for showing the cumulative CO₂ data and the explanation of how it could be influenced by the C additions and thus how the timing of samples could have resulted in the cumulative data being inaccurate. It is curious, and perhaps worth a comment in the manuscript, about the reason for large discrepancy for the DOC treatment.

Reviewer #3 (Remarks to the Author):

The major point the authors cannot respond to is the cumulative respiration. As these data underestimate loss of C, they seem not to be catching all effects of substrate addition. This is most probably due to the fact that CO₂ measurements have not been done insufficient density to allow to use the data for mass balance. This is a deficit in the data set. But as pointed out by the authors, in this case they can directly assess the amount of C in their samples, as they were initially C-free. All other points raised have been clarified.

“Direct evidence for microbial-derived soil organic matter formation and its ecophysiological controls”

We thank the reviewers for recognizing the novelty of this work and for helping to strengthen this MS. Our responses are provided below each comment or suggestion.

Reviewers' Comments

Reviewer #2 (Remarks to the Author):

I found this to be an exceptionally clearly written manuscript that describes a simple, elegant experiment to test the importance of microbial growth (and death) on the generation of soil organic matter.

Thank you for your enthusiasm for the manuscript and helping us improve it through the revision process. We have carefully considered your suggestions.

One glaring omission is the comparison of this study to other recent work that examines the same idea using artificial soils (Hemkemeyer et al. (2014) FEMS Microbiol Ecol 90: 770-782; Vogel et al. (2014) Soil Biology & Biochemistry 79: 57-67; and the references cited in these relating to prior artificial soil studies).

We have now included references to previous work using artificial soils in the Discussion section (L: 228-231 and 277-278).

l. 56-Although currently in vogue, the funneling of plant input through the microbial biomass is really not "new". Jenkinson (1977) famously stated that the microbial biomass is the needle through which all plant material much pass. Similarly, studies in the 60s and 70s showed that newly synthesized microbial biomass was found in the then-accepted solubility fractions used to describe soil organic matter (J.P. Martin, Roger Knowles, etc.). Like many ideas in science, this one is spiraling around again, albeit with better techniques, more clever experiments, etc. I would agree that it has been popularized and is the current favorite paradigm. We agree and have clarified this in the introduction and provided some historical context of earlier references to this idea (L 54-56). As you point out, this idea has gained significant traction in recent years along with greater recognition that the extent of direct microbial contributions to SOM may have previously been underestimated (in other words SOM does not just consist of the bits of plant residues leftover after decomposition). Here we expand on this idea by providing direct evidence (previously lacking) that substantial amounts of microbial inputs contribute to SOM formation and by demonstrating the relationships between microbial community and the amount of microbial inputs and subsequently SOM formation.

l. 197-My preference would be to not introduce RA as an abbreviation. We have reverted to unabbreviated form throughout.

l. 146-PLFA do give a broad fingerprint and have fungal and bacterial markers; however, it would have been desirable to have some deeper amplicon sequencing done on these samples to see if key taxa might be identified. (I realize that asking more is not quite "fair", so this should not be considered a major flaw.) Absolutely, it would be a valuable contribution if we could link specific taxa with our observations in physiology and SOM formation. However, for the purposes of this MS we feel that even detecting broad differences in

community assembly supports our conclusion that SOM formation is affected by differences in community composition and physiology. In fact, despite its drawbacks, we believe PLFA is a good tool for linking broad differences in communities, physiologies, and SOM chemistry and formation rates.

l. 200-If I did my calculation correct, your mesocosms held 100 g "soil". It might be helpful to put this detail somewhere in the methods. Thank you. This information is now provided in the Methods section (L 332).

l. 201-Wasn't there also a destructive sampling at 18 mo? Yes. This was clarified in the Methods sections (L 351).

l. 207-So, no mixing in prior 15 mo, correct? Why not? Correct. During the 15 mo. incubation period no soil mixing occurred in order to better simulate an undisturbed soil environment and to minimize aggregate destruction. In order to facilitate uniform application of the substrates during this time in the absence of soil mixing, additions were made via syringe-injection (5 mL of 14000 ppm C) throughout the soil. A statement as such to provide reasoning for no mixing was added to the Methods section (L 354-56).

l. 224-"Shimadzu". Corrected

l. 258-How do these values compare to your mass balance approach (Table 1), which I think would be more compelling data? This comment for L258 referred to the section in the methods of the previous version related to our approach for estimating microbial CUE (now L405-406) and so we are not entirely sure what the reviewer is asking in regards to our mass balance. We have, however, revised Table 1 to more explicitly include a mass balance between total C inputs, C remaining and subsequently C lost as respiration. The total system C use efficiency can then be calculated simply by determining the amount of total C added relative to the amount of accumulated soil C. These values (or the SOC conversion efficiency) broadly correspond to the treatment CUE.

l. 281-"different" rather than "significant". Corrected

l. 339-"regular". Corrected

Reviewer #3 (Remarks to the Author):

The manuscript describes a study on soil organic matter formation by microbial communities in an incubation experiment. The manuscript is well written and the research questions are well-formulated and relevant in the field of soil science.

The experimental approach is good and the data presented here together with those included in the supplementary information show a thorough characterization of the relevant microbial and carbon degradation processes. If the concerns listed below are addressed, it would make a valuable contribution to our current knowledge.

Thank you for your enthusiasm and constructive criticism of our manuscript. We have considered your suggestions carefully.

Some of the choices in data representation and interpretation are unclear, or could be reasoned better. Furthermore, the discussion of the results lacks context of relevant previous studies that have been done on this topic, or with similar experimental approaches. In particular, similar incubation experiments on the effect of clay mineralogy on microbial community establishment (e.g. Ding et al. 2013, Babin et al. 2013) and OM turnover (e.g. Pronk et al. 2012 and 2013, Saidy et al. 2013, Kaiser et al., 2014) should be addressed.

We thank the reviewer for their useful insights here and have incorporated their suggestions for some caution in data interpretation and better integrating our results with previous work. Specifically, we have included relevant references to artificial soil studies at L 228-231 and 277-278 and also provided context for previously observed differences in SOM chemistry across clay minerals (L 275-278 and 284).

--It is confounding how the authors use the term "soil" in the manuscript. E.g. in line 36 it is mentioned that you incubated "initially C free soil" or in line 85 you mention "soil mecosms" (e.g. line 85). I propose to make it clear that not a soil was used, but a mixture of clay mineral and quartz sand. Thank you, we have changed all references to the soil and soil mesocosms to "model soils". We introduce this term followed by a description of the soil and believe that now, and with its consistent use, it should be clear.

--Data and data evaluation: Essential information is missing

1. What is the total amount of substrate-C and N added to the mineral components/mixtures (kaolinite, montmorillonite, sand)? The total amount was 46 mg C and 1.56 mg N g⁻¹ soil at the end of the incubation. We have added this to the methods (L 343-344) as well as the total amount of substrate-C added at each sampling time point to Table 1.

--What are the total amounts of CO₂ respired after 6, 9, 12, 15 and 18 months?

A detailed balance is needed. We opted not to include cumulative CO₂ respiration since the frequency of CO₂ efflux measurements was not designed to adequately estimate the total CO₂ lost over time. Rather, respiration measurements were intended to track community activity and stability throughout the incubation. For a mass balance, we felt that it would be more accurate- and a more direct approach- to use the balance of total C inputs and the amount of total soil C remaining to estimate the amount of total CO₂ lost. These data are provided in Table 1 and are also used to calculate a SOC conversion efficiency, which is essentially the amount of soil C at any given time relative to the total amount of added C. We hope the mass balance approach based on C inputs and remaining soil C that we've now provided in Table 1 adequately addresses your concern.

Not all "SOM" after 18 months (Fig.2) can be considered as microbial-derived. Some SOM may be the "left over" of your substrate additions e.g. substrates and Hoagland solution and DOC that reacted e.g. with minerals surfaces. We agree and have included more cautious language in discussing the amount of total SOM accumulated. We have also included additional statements in the Results and Discussion sections (L 163-166; 185; 240-256) that provide

compelling evidence that most of the total accumulated SOM is likely derived directly from microbial residues. For example, while it is possible that glucose or cellobiose could react with minerals and accumulate, we see a >80% reduction in the py-GC/MS glucose and cellobiose signatures. Moreover, we can be certain that all of the lipids, chitin, and proteins are not from the substrates themselves (with the exception of the DOC which has these constituent within the pure DOC. For that reason we intentionally do not discuss microbial-derived SOM from DOC). We have elaborated on these points within the Discussion but remain careful not to make any absolute conclusions regarding the amount of microbial-derived SOM.

--How can the authors be sure that it is all microbial necromass? What about the contribution of microbial extracellular polymers (EPS). This cannot be ruled out with the data provided and might be a relevant proportion of the OM accumulated. We absolutely agree and included all microbial materials (including EPS production) in our definition of necromass. Upon reflection, it does seem that necromass is more suited specifically to cellular materials and so we have changed our terminology to reflect this as “microbial residues”.

Consequently the values in Table 1 are maximum values for "microbial derived SOM" and for the "SOC-conversion efficiency" under the specific conditions.

I propose to discuss the absolute values for "microbial derived SOM" and for the "SOC-conversion efficiency" more cautiously. Agreed. We have changed much of the language to be less quantitative about absolute microbial-SOM but have also provided some discussion about the results that support the microbial origin of accumulated SOM (L 240-256) (see comment above). In regards to the SOM conversion efficiency, we do not qualify that as being directly derived from microbes only that it represents the relative amount of SOC that accumulated compared to total C inputs.

To evaluate the SOM conversion efficiency it would be very helpful to present a balance that includes C losses by respiration. The mass of C lost can now be estimated directly from the revised Table 1 by subtracting C inputs from C remaining. Similarly, the SOC conversion efficiency is the amount of C remaining relative to total C inputs so it should be relatively straight forward for a reader to determine that, for example, with a conversion efficiency of 0.3 that 70% was lost as CO₂. While we could include these differences directly we felt that it would make the table far too large. By now having the C input rates in the revised table, the mass balance should be easily detected.

2. CO₂ mineralisation rates presented in Extended Data Figure 1 can also be used to discuss the biological stability of the substrate and SOM in the artificial soils. We have referenced the microbial activity (both the potential enzyme activity and respiration rates) in the discussion of microbial substrate utilization at L 185 and 240. Direct tests of both biological and chemical stability were performed and results are provided in Table 2 and presented at L 192-203.

3. Why didn't you calculate CUE directly by using microbial biomass C and CO₂ data that are presented in Extended Data Figure 1? Wouldn't this be a more direct proxy for the CUE of the microbial biomass that is actually active in the artificial soil amended with the specific substrates. There may be substrate recycling during the incubation but what does

it matter if original or recycled substrate is used by the microorganisms to obtain a proxy for the CUE of the actual microbial biomass? It is true that often times MBC and CO₂ are used to develop a metabolic quotient (qMBC) that is sometimes interpreted as a growth efficiency metric. However, this approach is really a measure of microbial activity, not efficiency. To determine efficiency, it is necessary to know the amount of substrate C that was used to create new biomass versus how much of that substrate was lost as CO₂. Thus, use of labeled substrates are a standard approach for estimating CUE (Geyer et al., 2016; Biogeochem.), though new approaches are currently being developed that don't require the addition of labeled substrate (¹⁸O method; Spohn et al., 2016 Soil Biol. Biochem.).

As stated in lines 75ff, CUE is influenced strongly by the substrate chemistry. For your CUE estimation, you used glutamic acid as substrate with a very narrow C/N ratio that is very different to the C/N ratio of the mixture of Hoagland's solution you used and your substrates or your DOC solution. You should discuss your CUE values against this background. Explain why you have chosen glutamic acid as substrate? What can be concluded e.g. from the result, that a microbial community that is adapted to mineralize syringol can use glutamic acid very efficiently? These are important points and we have provided some explanation at L 421-425, as well as included a Supplemental Note (Note 2). Briefly, this approach allows us to more directly compare CUE across treatments since substrate C allocation towards enzyme production is minimized and we can better approximate the efficiency of new biomass growth rather than substrate degradation efficiency.

Regarding the MBC and CO₂ respiration rates in Extended Data Figure 1, I would expect a lower CUE in the syringol treatment compared to more labile C-amendments. This tendency is contrary to your result (see line 163 ff) but in accordance to results from existing literature. Our data suggest that the microbial community in the syringol treatment becomes adept at utilizing syringol (the disappearance of the syringol chemical signature and the significant increase in enzymes and respiration after 6 mo of exposure). We suspect that syringol has selected for a community that is intrinsically more efficient at biomass synthesis following substrate uptake. In fact, once the syringol benzene ring is cleaved it yields more available energy than glucose. Regarding the MBC and CO₂ data please refer to comment above.

Line 120-127:

I agree with the authors that microbial processing is highly important for SOM formation. However, I question the reasoning the authors present here that their experiment demonstrates that the overriding effect is microbial processing. The material found after termination of the experiment can only be derived from microbial processing of the simple substrates added.

The incubations carried out here were done using relatively simple OM compounds. Even the DOC used was a switchgrass extract, not taking into account other sources of plant OM. It is not very surprising that the number of compound present increased after microbial processing for 18 months. We are unclear as to the issue raised here. The comment seems to be in agreement with our statements in the MS that we used simple substrates in the incubation and the remaining SOM is a product of microbial synthesis of these substrates. What we are intending to emphasize is that even with these simple C inputs we were still able to generate chemically diverse SOM resembling natural soils that typically receive an abundance of different

and chemically complex plant inputs. The origin of SOM's chemical diversity has often been attributed to the chemical diversity of plant inputs and the decomposition byproducts of microbial plant degradation, yet these results suggest that microbial inputs (metabolites, EPS, cytoplasmic materials etc.) may also have the chemical diversity necessary to generate the characteristic molecular diversity of most SOM. We agree that the DOC used is not completely representative of all plant inputs but generally DOC pools are a major contributor to microbially available C (Cotrufo et al., 2015 Nature GeoSci.; Kaiser and Kalbitz, 2012, SBB) and we are able to show that the initial DOC pre-incubation is indeed relatively more diverse (57 unique compounds-Fig. 1) compared to the sugars and syringol. Thus we believe the use of the DOC is a suitable natural analog for a more chemically diverse plant input. Certainly, model systems such as these have their limitation but they provide us with direct approaches for understanding the real-time development of microbial inputs to SOM and the interactive effects of substrate, mineralogy, and microbial communities on SOM formation.

It would also be interesting if the authors compared the OM compositions of their samples after incubation with that of microbial biomass. This may give more indication on the type of compounds preserved in soil. We have provided an additional table in Supplemental (Table 2) which compares the most abundant compounds across all treatments to a natural soil fungal biomass sample.

Lines 54-58:

How can this be a "new pathway of SOM formation" when the input composition of microbial necromass to SOM formation is described in detail in a review in 2002 (Kogel-Knabner, 2002)? Despite previous recognition that microbial residues contribute to SOM (we have included references to this L 54-55), there is very little experimental evidence that directly supports this, and the extent of microbial contributions to SOM (relative to plant-derived SOM) has not been adequately characterized. Much of our current understanding is based on correlative evidence of C/N ratios, higher ¹³C in older carbon, and select biomarkers. Indeed, in Kogel-Knabner et al.'s valuable 2002 review of the structural components of SOM, they state that the importance of fungal and bacterial constituents have "yet to be investigated". Our work expands on previous work in this field by demonstrating that a *large* portion of SOM is formed from microbial residues derived from simple substrate and that this SOM is both chemically diverse and biologically and chemically stable. Our results further advance the current theory that microbial residues generate an abundance of SOM by identifying some of the microbial community characteristics that might favor the production of microbial-derived SOM.

Line 138-140:

The differences between clay mineral composition are quickly discounted. However, looking at figure 2 there seem to be consistently higher aromatics contents in the montmorillonite treatment. There also seem to be differences in MBC and CO₂ (SI 1) and the NMDS points of the SOM chemistry cluster separately (SI 3). It would be valuable if the authors address these differences in the text. This is a good suggestion and we have included the observed differences in the chemistry between the mineralogies (as well as the microbial community), followed by a new section discussing these differences (L 174-177; 209-212; 270-286).

Line 158-170 and figure 3

This reasoning here is difficult to follow and seems somewhat circular. If fungal relative abundance correlates to SOC and CUE, it is a given that SOC also correlates to CUE (figure 3). Furthermore, the relative abundance of fungi, lipids and CUE are clearly not independent variables in this experiment. Particularly, since the difference seem to be dominated by the syringol treatment. We have used standard multivariate approaches (NMDS) here to demonstrate potential drivers of SOC accumulation and to characterize the relationships (Pearson's correlation) that may influence SOC concentrations. For the purposes of demonstrating the important relationships between CUE, SOC and fungal abundances we have chosen to highlight this graphically and to specifically show the strength of the individual relationships (r^2 value). We have included in the Supplemental section the correlations between these variables when syringol is not included in the analyses to demonstrate that the relationships we discuss are not primarily driven by one treatment, but persist even within sugar treatment experimental units (Supplemental Fig. 4). **How much of this can be explained by a higher capability of fungi to degrade the aromatic ring structure of syringol versus bacteria?**

We agree that the microbial community in the syringol-treated soils has likely become adapted to breaking down syringol. It is an interesting question as to how this occurs and there are several possible explanations (e.g. increased production of specific enzymes, community compositional differences) that we can only speculate on, though we believe does not necessarily enhance our discussion of results.

Line 170: "...characteristics that are highly sensitive to land use and climate change" this statement seems to come out of nowhere. This study seems to be more about the type of OM input than about climate change. We have expanded on how our results connect to more general aspects of soil C dynamics in relation to global change. This, we believe, highlights the relevancy of our results within the broader context of future changes in soil C and how we can improve upon predictions and its management (L 66-75 and 317-325).

Minor comments

- **Figure 1: I am not sure this provides useful information for the reader.** Feedback from presenting these images at scientific meetings indicates that audiences find them insightful in terms of visually demonstrating the qualitative differences in the model soils as they develop over time. Thus, we have chosen to keep the images, but have imbedded them within Fig. 1.
- **Figure 2: Should it be called lignin-derivatives if not lignin but specific compounds (i.e. syringol) were added?** The pyGCMS analysis retrieves hundreds of unique compounds that are grouped into classes according to their origin. There are typically several compounds that "originate" from lignin but are not necessarily *still* lignin. For example, syringol itself isn't lignin but a monomer that constitutes its macromolecular structure. Thus we use the class "lignin derivative" to indicate its origin and which encompasses several potentially identified specific compounds that fall into this class.
- **Figure 2: Improve contrast. The difference between lignin derivatives and chitin, and phenolics and proteins is difficult to see when printed.** Thank you. We have improved the clarity of the figure especially between compound classes and hope that you will find it easier to read.

“Direct evidence for microbial-derived soil organic matter formation and its ecophysiological controls”

We are pleased that we were able to satisfy most of the reviewers' initial requests and are grateful to have the opportunity to address their remaining concerns. Our responses are provided below for each comment or suggestion.

Reviewers' Comments

Review response comments:

The authors have responded thoroughly to the comments of the two reviewers. I am satisfied with their response except in one area: use of cumulative CO₂ data to close the mass balance. Despite the argument provided by the authors, the sampling intensity (about 54 times during about 540 days) is reasonably intense and I think sufficient to at least calculate a ballpark estimate of the cumulative amount of CO₂ produced at the times that soils were destructively sampled. I think this should be done and the data added to Table 1. It is not important that the balance close perfectly, but if it is off by more than 10% or so, and especially if there is a consistent over- or under-estimate, then it may suggest that the data are not as sound as they should be.

We have given this careful consideration but continue to believe including these data does not improve the manuscript and in some ways could be misleading. The CO₂ cumulative data are provided below for your consideration and review.

We would like to emphasize that the intent of measuring CO₂ over the course of the experiment was to ensure that the microbial community was active and responsive to substrate additions. Certainly, it is not uncommon to use cumulative respiration for estimating a C mass balance, but when conditions allow, using soil C stocks is a much better integrator of mass C balance than CO₂ fluxes, which are ephemeral and temperamental. We are fortunate in this experiment to be able to do this since we began with C-free soils and carefully accounted for all C inputs.

While we do have ‘reasonably intense’ sampling for CO₂, we believe that, in our systems, to conduct a mass balance using these data requires more measurements. This is especially true in our soil systems, in which the weekly substrate inputs resulted in intense fluctuations in respiration between additions. The magnitude of the efflux is far greater immediately following additions relative to 7 days post-C additions. These fluctuations are demonstrated in Figure 1 below. During the initial first few months and towards the final months of the experiment we characterized these fluctuations by taking CO₂ measurements both pre- and post-substrate additions (again, to establish that the microbial community was actively responding to the substrates, and if it was active or not before the next addition) but do not have these data for about 8 months of the experiment. Thus, we believe we cannot adequately extrapolate cumulative respiration based on point measurements which vary significantly depending on when the measurements were collected.

Some of the cumulative respiration data is consistent with our calculations for C lost based on our SOC mass balance approach, though in some treatments cumulative respiration significantly underestimates C lost (Table 1 below). There can be many interpretations to explain these data due to microbial community differences and the timing and magnitude of respiration responses to substrate inputs that are interesting to

consider but are not relevant to the main objectives of this study and ultimately we believe do not influence the interpretation of data in our manuscript.

As a compromise, we have included the cumulative respiration data in the Supplemental section as Table 4. We hope given these explanations and the data provided here that you will agree that, in our case, the cumulative respiration data may not only be an inaccurate measure of soil C lost but could also be misleading to the reader and does not necessarily provide additional value to the manuscript.

	Glucose	Cellobiose	Syringol	DOC
	mg C g-soil			
Kaolinite				
Cumulative respiration	24.8	24.4	.	8.7
SOC mass balance approach	38.3	38	.	36.5
CV	0.303	0.308	.	0.87
Montmorillonite				
Cumulative respiration	44.1	43	34.9	10.9
SOC mass balance approach	38.54	38.92	33.79	35.2
CV	0.095	0.07	0.023	0.745
Total C added	46.9			

Table 1. Comparison of soil carbon lost based on cumulative respiration and a mass balance approach from total C inputs and total C remaining and the coefficient of variation (CV) of the two approaches.

Specific points:

1. 118-136—This is a stylistic comment, but the first two paragraphs of the results are really a recapitulation of the methods. But maybe it has to be this way because of the structure of sections in this journal.

Yes, we arranged it this way because of the journal structure and realize it may not be conventional but have also found in previous *Nature Communications* publications (e.g. doi:10.1038/ncomms11965, doi:10.1038/ncomms8628) similar approaches for briefly integrating Methods into the Results.

We found that without a clear understanding of the experimental setup upfront that it makes it more challenging for the reader to follow the results and we also wanted to provide some context and rationale for the experimental set-up (e.g. the choice of substrates,

the initial conditions of the soil, and the sampling). But we agree that technically this section is not Results material. To improve the flow we have reduced the details of this section and moved most of the sentences related to the rationale of the experimental design to the last paragraph of the introduction.

l. 162—Is this really “chitin” or would bacterial cell wall materials also be lumped into this category? **Yes. This is based on molecular biomarkers that have been previously identified for py-GC/MC compounds available in NIST and the compound library previously developed by the co-author, Grandy (2-Acetylfuran; 3-acetamidofuran; Acetamide, N-(2,4-dihydroxyphenyl)-; Acetamide, N-hydroxy). The power of py-GS/MS is it allows us to examine individual molecules within broader functional groups. Bacterial cell wall constituents (as well as fungal) are included more broadly in our polysaccharide group.**

l. 352—“months”

Corrected

l. 432—The statistical analysis description suggests that interactions between substrate and clay types were calculated, but it is unclear if they were ever significant. The presentation of the data in the tables and figures is not clear enough to determine if there were interactions or not, although given the unbalanced design they seem to be presented as if there were interactions. **Yes, we did determine interactive effects but we have, for simplicity, focused on demonstrating the main effects in the text and figure designs. In some instances we have made references (e.g L187, L198) to interactive effects where, for example, differences in mineralogy only occur for a particular substrate. We have added to the Supplemental section, ANOVA tables (Table 3) for both the two-way analyses which don't include the syringol treatment, as well as the one-way analyses for examining main effect of substrate within mineral treatments. We believe these summary data will provide additional clarity regarding the analyses conducted as well as when main effects and interactive effects were significant. Language was also clarified in the Methods L434:437.**

Table 1—Show all data to hundredths position.

All data is now reported to the hundredth position.

Also, the use of significance letters is confusing. Usually the alphabetical order follows either the increase (or decrease) of the values, but this is inconsistent (e.g., 6 mo, substrate comparisons for montmorillonite treatment—why does C represent a number between those associated with A or B?). This point about significance letters is also the case in the other tables and figures.

Thank you- this was an oversight and we have now changed significance letters to correspond with values following increasing order in all tables and figures.

Reviewer #3 (Remarks to the Author):

The authors have positively responded to the points raised in the review and amended the manuscript. They have now also included further references that put their work now in the correct context with respect to previous attempts to investigate the microbial contribution to soil

organic matter formation.

One point still remains: The term “microbial residues” used now, instead of “microbial necromass” is misleading and often the terms are used in the literature as synonym. It is thus necessary to give a clear definition, what the term encompasses. It is also necessary to state that both necromass *sensu strictu* as well as extracellular polymeric substances (often called EPS) cannot be differentiated with the present experiment. EPS are released from living cells and it should be made clear what materials are dealt with in the paper.

We agree that dead cellular material cannot be differentiated from other microbial materials (e.g EPS) with our experiment. In our experiment we intended to capture all microbial materials since it remains unknown which microbial constituents make the most significant contributions to SOM. To clarify this, we have removed “*dead*” from the abstract L27 and included a statement specifically identifying the constituents we consider in the interpretation of our microbial residues L117-18 in the last paragraph in the Intro. Moreover, L60-61 in the Intro’s first paragraph and L:245 and 255 in the discussion include “excretions” in our description of materials associated with of microbial-SOM.

[Reviewer comment: “Thank you for showing the cumulative CO₂ data and the explanation of how it could be influenced by the C additions and thus how the timing of samples could have resulted in the cumulative data being inaccurate. It is curious, and perhaps worth a comment in the manuscript, about the reason for large discrepancy for the DOC treatment.”]

[Response: We have included a few sentences (Lines 181-189) regarding cumulative respiration to reflect the reviewers’ comments provided during the review process. Specifically, as requested by reviewer #2, we addressed the discrepancy with the DOC treated soils.]